# Antimicrobial Activity of Spanish Propolis against *Listeria monocytogenes* and Other *Listeria* Strains

**DOI:** 10.3390/microorganisms11061429

**Published:** 2023-05-29

**Authors:** Eugenia Rendueles, Elba Mauriz, Javier Sanz-Gómez, Félix Adanero-Jorge, Camino García-Fernandez

**Affiliations:** 1Institute of Food Science and Technology (ICTAL), La Serna 58, 24007 León, Spain; jjsang@unileon.es (J.S.-G.); fadaj@unileon.es (F.A.-J.); mc.garcia@unileon.es (C.G.-F.); 2ALINS, Food Nutrition and Safety Investigation Group, Universidad de León (ICTAL), La Serna 58, 24007 León, Spain

**Keywords:** *Listeria*, antimicrobial activity, propolis, food pathogens, physicochemical composition, phenolic compounds, bioactive properties

## Abstract

The outbreaks of *Listeria* associated with food consumption are increasing worldwide concurrently with public concern about the need for natural growth inhibitors. In this context, propolis seems to be a promising bioactive product collected by honeybees, due to its antimicrobial activity against different food pathogens. This study aims to evaluate the efficacy of hydroalcoholic propolis extracts for controlling *Listeria* under several pH conditions. The physicochemical properties (wax, resins, ashes, impurities), the bioactive compounds (phenolic and flavonoid content), and the antimicrobial activity of 31 propolis samples collected from the half North of Spain were determined. Results showed similar trends in the physicochemical composition and bioactive properties, regardless of the harvesting area. Non-limiting pH conditions (7.04, 6.01, 5.01) in 11 *Listeria* strains (5 from collection and 6 wild strains from meat products) exhibited MICs (Minimum inhibition concentration) and MBCs (Minimum bactericidal concentration) ranging from 39.09 to 625 μg/mL. The antibacterial activity increased under acidic pH conditions, showing a synergistic effect at pH = 5.01 (*p* < 0.05). These findings suggest the potential of Spanish propolis as a natural antibacterial inhibitor to control *Listeria* growth in food products.

## 1. Introduction

Despite advances in the food industry in technology, safety, and hygiene throughout the food chain, *Listeria monocytogenes* remains one of the most relevant food pathogens. The latest report issued by EFSA in December 2022 [1], which includes data on both food outbreaks and the incidence of infections caused by this microorganism during the previous year, refers to the foods involved and the countries of the European Union in which these cases have occurred. According to this report, EFSA has declared more than 2000 cases of listeriosis associated with the human population, involving 23 outbreaks and almost 10% of the samples registered for this gram-positive pathogen. The number of confirmed cases over the last five years remains more or less stable, although there has been an increase in the number of outbreaks reported in the twenty-seven member countries.

A significant increase is in the number of samples that member countries have reported in the different ready-to-eat (RTE) products responsible for most of the outbreaks of *L. monocytogenes* associated with the consumption of these products. In the last year, the number of sample units in meat, dairy, and fish products increased by almost 50%, highlighting the effort and importance of *L. monocytogenes* control.

Together with the application of the current legislation in the inspection and control of the incidence of *L. monocytogenes* in RTE, the agri-food industry has implemented research in the search for and use of different technological treatments, additives, and supplements that contribute to the safe control of the growth and multiplication of this pathogen [2]. In addition, natural substances that contribute not only to the improvement of safety, but also to the concept of Clean Labels, which is so appreciated by the consumer, have been advocated lately.

In this context, propolis is one of the hive products that stinging honeybees (*Apis mellifera*) and stingless honeybees (*Melliponas*) produce from buds, bark, leaves, and exudates of trees and plants [3,4,5]. The composition of this sticky, resinous substance consisting of waxes, resins, pollen, water, and some impurities varies greatly depending on the time of year, the geographical location of the hives, and, of course, the flora.

The organic composition of propolis is also highly variable, the main components of which are acids, polyphenols, terpenes, esters, and minerals. Beneath them, polyphenols, including flavones (apigenin, luteolin), flavonols (quercetin, kaempferol), flavanones (naringin), and flavanols (epicatechin) [6], are particularly relevant due to their proven biological activity. This bioactivity translates into antioxidant, anti-inflammatory, antimicrobial, and antitumor effects, thus enhancing their role as immunomodulators and biomarkers [7].

The therapeutic effects of propolis extracts for different pathologies in humans, such as oral mucosa and upper respiratory tract infections, persist for millennia in different cultures and societies. Their most recent applications are perhaps in the food industry as a natural substance for the pathogenic control of microorganisms of interest.

Moreover, beekeeping is considered a sustainable and resilient agricultural and livestock farming activity, which helps to maintain the environment and keep the population in rural areas. For instance, beekeepers implement propolis collection management practices using nets and grids to obtain this product more efficiently [8,9].

The bactericidal and bacteriostatic effect of the different polyphenols and flavonoids that comprise the resin fraction on gram-positive microorganisms is well known [7]. Propolis bioactivity on *Listeria* spp. and *L. monocytogenes* (bacillus g(+)) has been studied by some authors in some meat, dairy, vegetable, and fruit products [10,11,12]. Given the organoleptic characteristics of propolis (potent aroma and intense color), it is necessary to determine low effective doses in its application to food products [3]. Similarly, the *Listeria* inhibitory effect, without interfering with its aroma and color properties, is crucial for guaranteeing safety in food products [13]. The effect of pH on the survival of *L. monocytogenes* under the bactericidal and bacteriostatic properties of the bioactive propolis compounds plays a pivotal role in allowing *Listeria* growth [14]. Thus, values higher than 4.0 have been established by the current legislation (European Directive 2073/2005) for ready-to-eat products.

Despite these considerations, only a few studies address the pH effect on the growth of different *Listeria* strains [12,15]. Moreover, research on the synergistic bioactivity between the ingredients or additives in food matrices is still scarce. Therefore, this study aims to contribute to the potential application in the food industry of Spanish propolis as a control of *Listeria* in ready-to-eat products under different pH conditions. This would thus provide an alternative to the food industry by labeling its products as Clean Label [16], since propolis is a supplement of natural origin, as well as revaluing the economic value for beekeepers.

## 2. Materials and Methods

### 2.1. Materials and Reagents

All the extraction assays used Ethanol from Panreac and Methanol (Labkem, Barcelona, Spain). The preparation of the calibration graph involved the following products from different trading houses: Pinocembrin, Med Chem Express and Galangin, Med Chem Express, both from Sweden and Quercetin, Sigma Aldrich (St. Louis, MO, USA). The spectrophotometric measurements required a Beckman DU-7400 Spectrophotometer. Culture media for microbiological determinations were Mueller Hinton Broth (MHB) and Mueller Hinton Agar (MHA), both from Bioser (Barcelona, Spain). Antimicrobial Susceptibility Test Discs and Resazurin (7-hydroxy-3H-phenoxazine-3-one-10-oxide) from Oxoid (Basingstoke, UK) were also employed. Petri Dishes and V-bottom microtiter plates were from Deltalab (Barcelona, Spain). The adhesive film for Culture Plates Porous was from VWR (Radnor, PA, USA).

### 2.2. Groups of Propolis and Sample Collection

A total of 31 samples of propolis were collected from several geographical areas in Spain throughout 2019 and 2020. Beekeepers from different regions of Spain voluntarily participated in the study, collecting propolis samples according to their usual management practices. Macroscopically, differences were observed among raw samples, including texture, color, and compactness degree. Figure 1 shows the geographical origins of the collected propolis, grouping some of them in the N-W region of the Iberian Peninsula, most of them in the province of León, Zamora, Palencia, Asturias, and Madrid (18) Zone I, Galicia (5) Zone IV, Catalonia (2) Zone II, and Navarra (2) Zone III. The areas of origin of the propolis samples were classified according to the administrative limits corresponding to the Autonomous Communities of the Kingdom of Spain (Appendix A). Propolis was stored in freezing conditions (−20 °C) from collection to analysis.

### 2.3. Preparation of Propolis Extracts

Propolis extraction of pulverized raw propolis (10 g) comprised the addition of 600 mL of 70% hydroalcoholic solution. The extraction was carried out in two stages of 24 h, each of which involved shaking, at a controlled temperature of 20 °C, the 10 g of propolis sample, first in 300 mL of 70% ethanol, and after filtering a new extraction under the same conditions of shaking, time, and temperature [4,17]. Ethanol extracts of propolis (EEP) were kept at a refrigerated temperature (4 °C) from that moment and throughout the procedure to keep their properties intact [17].

### 2.4. Characterization of Propolis Samples

EEPs were analyzed to determine the main components responsible for their quality, bioactivity, and chemical properties. The characterization of the selected propolis samples considered their proximal composition regardless of their botanical origin, including resin, water, wax, impurity, and ash content [4,17].

### 2.5. Determination of Total Polyphenols Content

Total polyphenol content (PTC) was analyzed according to Folin–Ciocalteu’s method (as modified by Bankova et al. [4]). The calibration graph was prepared with standard methanolic solutions of a mixture of pinocembrin-galangin at a ratio of 2:1 (*w*/*w*) in a 25–300 μg/mL range. Absorbance measurements were performed in triplicate at 760 nm using a UV-vis spectrophotometer, and the results obtained were expressed as grams of pinocembrin-galangin equivalents per 100 g of raw propolis (%PGE) [4,18,19].

### 2.6. Determination of Total Flavonoids, Flavone, and Flavolnol Content

Total Flavonoid Content (TFC) was estimated according to Woisky and Salatino, using quercetin as a standard and expressing the results obtained as grams of quercetin equivalents per 100 g of raw propolis (%QE). Absorbance measurements at 415 nm after 40 min of incubation in the dark at room temperature against a blank were performed using a spectrophotometer [4,18]. Flavone and flavonol content was determined using the spectrophotometric method proposed by Bankova et al. [17] based on the reaction to form aluminum chloride complexes. The calibration line preparation involved a methanolic solution of galangin. Absorbance measurements were performed in triplicate at 425 nm using a UV-vis spectrophotometer. The results obtained were expressed as grams of galangin equivalents per 100 g of raw propolis (%GE) [4,18].

### 2.7. Antibacterial Activity

#### 2.7.1. Bacterial Strains

The development of this assay involved 11 strains of different origins. Five were collection strains (CECT): CECT 910 *L. innocua* ser. 6a, CECT 931 *L. grayii* serovar 4a (*sensu lato*) (*MIYAGUE*), *CECT 4032 L. monocytogenes* serovar 4b, CECT 934 *L. monocytogenes* serovar 4a, and CECT 935 *L. monocytogenes* serovar 4b. The wild strains were isolated from different meat products: sliced cecina (L74 *L. monocytogenes* (*Murray*), L75 *L. monocytogenes* (*Murray*)), cured chorizo (L10 *L. monocytogenes*), and sliced cooked meat products (L51 *L. monocytogenes* serovar DSM206007 1/2a serovar, L52 *L. monocytogenes* DSM206007 1/2a serovar, L 30 *L. innocua* DSM2049T serovar 6a) (Appendix A).

#### 2.7.2. Antimicrobial Disc Susceptibility Test

The disc diffusion method was used to evaluate the antibacterial activity of EEP. Petri dishes were prepared with MHA and inoculated with *Listeria* culture in MHB before overnight incubation at 37 °C (2 × 10^8^ cfu/mL). Afterwards, eight peripheral sterile Whatman filter paper discs with a symmetrical arrangement template and a central one were placed in each plate. Twenty µL of each of the thirty-one EEPs were placed in the peripheral discs, using the same volume of 70% ethanol as a negative control. The growth of the inoculated strain was evaluated after incubation for 48 h at 37 °C, and the diameter of the inhibition generated around the discs was measured while using the diameter of the inhibition of the central disc of each plate as a control. The diameter zones, including the diameter of the disc, were recorded [4,20].

#### 2.7.3. Minimum Inhibitory Concentration (MICs) and Minimum Bactericidal Concentration (MBCs) Determination Assay

From the EEP and based on the results of the composition analysis and determination of active compounds, further dilutions ensured a range concentration of 2500 µg of raw propolis/mL in the most concentrated dilution. MIC determinations involved three pH values, non-limiting for *Listeria* growth. The pH of the MHB was adjusted post-sterilization. The selected pH values were 7.04, 6.01, and 5.01 to determine if the pH conditions modified the antimicrobial activity of the EEPs. V-bottom microtiter plates contained 50 µL of MHB in each cell. The MIC values determination, namely the minimum concentration that allows for visible microbial growth, was achieved through known serial dilutions, using the last row of each plate as a growth control without EEP. The propolis’ inoculated concentrations ranged from 2500 µg/mL to 39.0625 µg/mL in the most diluted one. Finally, 50 µL of the bacterial inoculums in MHB were included in each column and incubated at 37 °C for 48 h (all covered with breathable rayon film). The dilution tested was the one that achieved 1.2 × 10^5^ cfu/mL. Seeding was performed in a Petri dish containing MHA, with a multi-inoculator, which was incubated at 37 °C for 24 h, thus determining the MBC values [4]. In addition, complementary and parallel staining with Resazurin at 0.01% showed a redox reaction, indicating the presence of viable cells in the dilution by a color change (it is naturally blue and turns purple) [4,21]. Results were obtained after 4 h of incubation at 37 °C using the 3-pH values selected in the test.

### 2.8. Data Analysis

Continuous variables were expressed as mean values ± standard deviation (SD). The Kolmogorov–Smirnov test was applied to determine data normality. Differences between zone regions were analyzed using One-way ANOVA and Kruskal–Wallis tests for normal and non-normal distribution variables, respectively. The influence of pH on the MICs values was assessed with the Kruskal–Wallis test. The software package SPSS for Windows version v.26, (IBM SPSS, Inc., Chicago, IL, USA) was used for data analysis. A *p*-value of <0.05 was set as representing statistical significance for all analyses.

## 3. Results

### 3.1. Physicochemical Characterization

The analysis of propolis samples allowed for their classification according to the established standards. The content of soluble matter in 70% ethanol (resin fraction) was 3–0.9%. Impurities and water content showed values between 0.01 and 0.11% and 3 and 6% for impurities and moisture content, respectively. Higher wax content was found in zone III (39.04%), while the lowest wax content was found in zone IV (7.74%). Non-statistically significant differences were found between areas.

### 3.2. Bioactive Compounds

The analysis of the total resin content did not reveal statistically significant differences between different areas. Considering the two bioactive fractions of the soft extract of the study samples, no significant differences in TPC were observed between samples when the whole regions were considered. Neither TFC nor FFCT were found. Table 1 shows the total phenolic compounds, and the concentrations ranged between 26.65 ± 0.72% PGE in sample 16 and 78.54 ± 0.80% PGE in sample 4. The same observation was made for flavones and flavonol content, where the highest concentration was found in sample 4 (5.71 ± 1.07% GE), immediately followed by sample 3 (5.53 ± 0.28% GE) and sample 10 (5.45 ± 0.61% GE). In addition, the total flavonoid content has a concentration between 1.64 ± 0.04% PGE and 4.94 ± 0.05% PGE.

### 3.3. Antibacterial Activity

The tests carried out to perform a first screening of the antimicrobial activity of each EEP provided homogeneous information in terms of the results shown in Table 2. Both wild *Listeria* strains and CECT showed inhibition halos diameters ranging from 38 mm for L10 (*L. monocytogenes* isolated from a cured chorizo), which exhibited greater sensitivity to the action of the EEPs, to *L. innocua* CECT 910, whose lowest inhibition diameter was 5 mm.

All the propolis samples showed antibacterial activity against all the *Listeria* strains. MIC mean values were between 39.025 and 625.00 μg/mL (Table 3).

The anti-listerial activity increased as the pH diminished, displaying lower MIC mean values at pH 5.01 when considering either geographical zones or *Listeria* strains. Specifically, the lowest MIC mean values at pH 5.01 were found for L30 (102.96 ug/mL), while the MIC mean value between zones was 97.66 μg/mL for Zone III. At pH 7.04, the MICs were 196.57–253.59 μg/mL (Lm 931-L910); pH 6.01 from 162.45 to 211.69 μg/mL (Lm 934-Lm10), and the lowest mean values to pH 5.01 were from 114.68 to 156.25 μg/mL (Lm 934-L30) (Appendix A).

Moreover, differences in MIC mean values between pH 7.04 and 6.01 were not statistically significant for all *Listeria* strains, except for L10, which did not exhibit statistically significant differences between pH 6.01 and 5.01.

Although different strains of *Listeria (L. monocytogenes*, *L. grayii*, and *L. innocua*) were tested, the growth behavior was similar. It was only at pH 5.01 that the EEP samples from two geographical areas (Zone III and Zone IV) displayed statistically significant differences among *Listeria* species.

Similarly, MBC values displayed the lowest result from 115.69 μg/mL at pH 5.01, which was exhibited by Lm 4032, and the higher MBC value was 253.28 μg/mL at pH 7.04, which was shared by Lm 4032 and Lm 52.

## 4. Discussion

This study investigated the anti-listerial activity of Spanish propolis against different strains of *Listeria*. As far as we know, this is the first work that explores the effect of pH on different *Listeria* strains isolated from meat products, using simulated acidic pH conditions present in several RTE products. The propolis samples selected in this study come from a wide geographic area in the northern half of Spain, within the European or poplar-type, although showing climatic diversity depending on the geographical origin. The samples’ appearance was heterogeneous in terms of the compactness degree and color, owing to the different ways of collection. Most beekeepers placed a grid or mesh for collecting, while only one sample was obtained by scraping (Propolis 18). Additionally, harvesting occurred for two consecutive years (2019–2020).

Regarding the physicochemical composition, the heterogeneity of the chemical composition of propolis rendered standardization challenging [22], making a primordial determination of sample quality, including parameters such as wax, resins, ashes, and humidity. All 31 propolis samples collected from different harvesting areas showed values that comply with the standards established for European propolis, as seen in some published works on Spanish and Portuguese propolis [23,24]. The limits of acceptable values following the protocols and specifications [4] listed in the IHC (International Honey Commission) are as follows: a minimum of 45% resins; a maximum of 6% impurities, a maximum of 8% water content, and a maximum ash content of 5%.

The total of the 31 propolis studied presented in their resinous fraction contents in polyphenols, flavonoids, and flavones that correspond to Popova et al.’s [18] previous studies for poplar-type, with a minimum of 21% in total polyphenols content (CPT) and a minimum of 4% in flavones and flavonol content (CFFT). Despite the geographical origin [25], climate, botany around the hive, year of collection [26,27], and management, among other factors, the wide range of the results obtained in the characterization analysis of the samples did not show statistically significant differences between areas. Concerning the bioactive contents, previous works have characterized European propolis (poplar-type) as rich in flavones, flavonones, and flavones [28], while the Russian and Brazilian types present other bioactive substances (diterpenoids, isoflavones, and other flavonols) [29,30]. Nevertheless, all the samples presented homogeneous values that correspond to those already shown in previous works [3,31]. TPC content (some higher than 70% PGE) and TFC (most of them higher than 4%) are higher than that reported for Portuguese propolis [31,32]. The flavonoid content did not change between regions, although there were different vegetation sources around the hive. Regarding the flavanones and flavonols subclass, both agree with the previous characterization of the poplar-type of propolis [18]. Numerous studies attribute the propolis’ bioactive properties (antibacterial, anti-inflammatory, antioxidant [33], biomarker [34], and immunomodulator) to the compounds present in the resinous fraction. Therefore, polyphenols and flavonoids have been described and used in the pharmaceutical industry and the medical field for their bactericidal and bacteriostatic effects [35].

Despite the broad knowledge in this respect, few previous works have reported the control effect exerted by propolis on *Listeria* [36], particularly on gram-positive microorganisms and some molds [31,37,38,39]. In the present work, involving 31 EEP against 11 strains of *Listeria* with different origins, the inhibition screening carried out with the disc diffusion method [40] showed that both wild and collected *Listeria* strains were sensitive to all the EEP studied.

The antibacterial effect of propolis on strains of *Listeria* demonstrated that all EEP inhibit this foodborne pathogen. Since extracted propolis samples contained ethanol, 70% ethanol was used as a control to evaluate its potential inhibition effect. Growth inhibition against the tested *Listeria* to the ethanol was not observed, thus suggesting that the antimicrobial effect was due only to propolis.

The ethanol extracts propolis showed an inhibition zone diameter higher than 12 mm for all the *Listeria* strains evaluated [40]. The results of minimum inhibitory concentration (MIC) found by the 31 propolis ethanol extracts showed high inhibitory effects against *L. monocytogenes*, *L. grayii*, *and L. innocua*, both the collection strains and wild ones isolated from meat products. As expected, the propolis samples showed bactericidal and bacteriostatic activity against the different strains of *Listeria* spp. and *L. monocytogenes*, independently of the EEP tested. The MIC values found when confronted with the 31 propolis against *Listeria* showed lower values—higher inhibition power—than previously published work with propolis of very different geographical origin than those presented in this study, demonstrating that the bactericidal effect is superior (MIC 39–625 µg/mL).

Only a few works have reported the propolis bioeffect against *Listeria*. Almost all of them compared results between gram-positive and gram-negative bacteria [25,41,42,43,44] or studied *Listeria* in Ready-to-Eat (RTE) products, in addition to different conditions and components [11,16,31,45,46,47,48]. Previous studies against gram-positive microorganisms [49] with European propolis revealed MIC values ranging from 80 to 5000 µg/mL, while Polish propolis (400–8000 µg/mL) [41,42] revealed higher MIC values, referred to as *L. monocytogenes* (1560–6250 µg/mL). In our study, all the propolis assayed exerted a high inhibitory effect against the whole *Listeria* strains tested. The strain origin (wild/collection) did not show significant differences in sensibility to EEP. The mean MIC value for pH 7.04 was 265 µg/mL. Although some of the independent MIC values obtained when we tested pH 6.01 were very low (199 µg/mL), the acidic assay conditions, pH 5.01, revealed a mean MIC value of 134 µg/mL, thus suggesting a potential synergistic effect between pH activity and the action EEP for the total *Listeria* strains tested, including both wild and collection types.

The antimicrobial effects of propolis may be influenced by many factors, such as the propolis origin, extract preparation, and chemical composition, also exhibiting considerable geographic differences. Similarly, the antimicrobial effect (expressed as MIC) of tested EEP was lower than 625 µg/mL, making this concentration compatible with their use in the food industry, as they do not modify the organoleptic characteristics of the food matrix, as reported by other natural substances in previous works [13,50]. For instance, flavor is one of the key components of propolis characterization and it should be considered when it is applied to RTE products. Nevertheless, the imitation of the pH conditions present in many RTE foods and the study of the propolis effect against 11 *Listeria* strains has not been previously described. The simulation of the pH values in some of the conditions for meat products and the bacterial damage observed in the most acidic pH could be extremely valuable for the food industry as an alternative way to control this food pathogen.

The low MIC found in our propolis samples evaluated against *L. monocytogenes* and other *Listeria* are compatible with its potential use in the industry for food safety control. Despite these inherent advantages, this study has some limitations. For instance, the sample size could be homogenized within zones, considering the geobotanical characteristics. Although we provide specific results on total flavonoids, several subclasses could be further analyzed.

## 5. Conclusions

The control of *Listeria* in RTE products through natural bioactive compounds is of utmost importance for ensuring food safety. This study demonstrates the utility of the propolis extract from the northern half of Spain as a potent inhibitor of *Listeria* growth. Particularly, the studies of different pH conditions revealed that the synergistic effect of acidic values increased the antibacterial properties of propolis samples. These findings could contribute to defining future strategies for the bio-control of *Listeria* growth in RTE and meat products using specific pH conditions. Therefore, this investigation supports the promising integration of natural and sustainable products as efficient controllers of pathogens in the food industry. Further research is necessary to enforce the inclusion of propolis as a Clean Label product to preserve the shelf life while protecting consumers’ health.

## Figures and Tables

**Figure 1 microorganisms-11-01429-f001:**
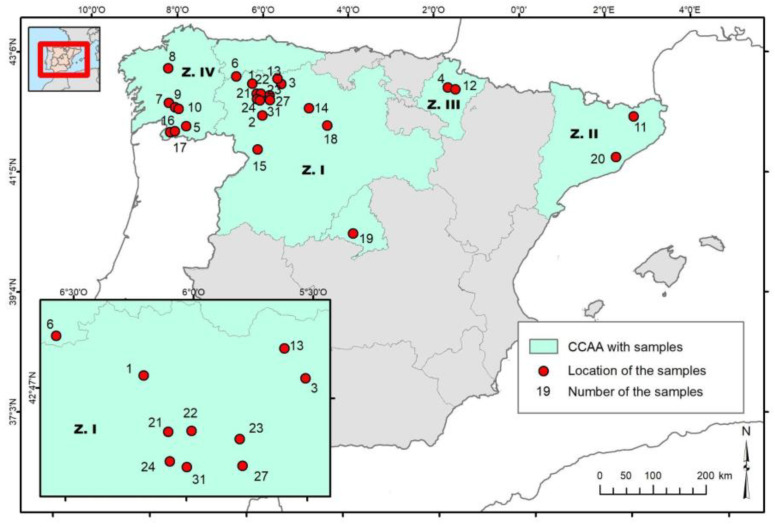
Representation of the geographical origin of the propolis samples.

**Table 1 microorganisms-11-01429-t001:** Physico-chemical characteristics of propolis extracts measured according to Resins Content (RC), Total Phenolic Compounds (TPC), Total Flavonoids Content (TFC), and Total Flavone and Flavonol Content (TFFC) grams in 100 g Raw Propolis Sample.

Zone Code	Geographical Origin	Propolis	RC	TPC (%PGE)	TFC (%QE)	TFFC (%GE)
I	Castilla y León and adjacent	1	60.52 ± 2.66	55.60 ± 1.20	4.15 ± 0.43	4.98 ± 0.25
		2	65.20 ± 2.89	60.72 ± 0.96	4.88 ± 0.09	5.26 ± 0.13
		3	55.05 ± 1.89	50.04 ± 0.96	3.55 ± 0.14	5.53 ± 0.28
		6	50.73 ± 6.20	29.66 ± 1.83	2.04 ± 0.05	4.55 ± 0.82
		13	66.91 ± 5.25	33.70 ± 2.86	4.69 ± 0.14	4.39 ± 0.15
		14	61.08 ± 3.78	29.50 ± 1.02	3.50 ± 0.04	3.70 ± 0.76
		15	56.90 ± 1.33	33.09 ± 0.21	3.00 ± 0.03	3.35 ± 1.18
		18	66.82 ± 1.69	34.60 ± 0.72	3.84 ± 0.08	3.52 ± 0.26
		19	47.66 ± 1.46	42.13 ± 0.33	3.84 ± 0.55	4.34 ± 0.24
		21	76.26 ± 1.80	56.94 ± 0.00	4.66 ± 0.45	4.17 ± 0.31
		22	74.70 ± 0.86	71.64 ± 0.42	4.68 ± 0.02	4.10 ± 0.03
		23	73.88 ± 1.05	39.31 ± 1.51	4.13 ± 0.63	4.20 ± 0.23
		24	72.62 ± 1.73	37.31 ± 1.93	4.95 ± 0.36	4.33 ± 0.30
		25	72.61 ± 5.08	45.14 ± 0.11	4.46 ± 0.27	4.36 ± 0.03
		26	72.51 ± 1.56	36.85 ± 0.64	4.32 ± 0.26	4.18 ± 0.12
		27	68.73 ± 1.22	46.74 ± 1.51	4.39 ± 0.37	4.03 ± 0.31
		28	64.99 ± 2.44	38.63 ± 0.30	3.31 ± 0.01	3.78 ± 0.28
		29	65.19 ± 5.10	31.66 ± 0.20	3.45 ± 0.36	3.79 ± 0.13
		30	72.09 ± 5.55	29.40 ± 1.07	3.11 ± 0.25	3.90 ± 0.56
		31	67.87 ± 1.70	31.31 ± 0.10	3.84 ± 0.08	4.13 ± 0.56
		Mean And DS	65.62 ± 8.11	41.80 ± 13.22	3.94 ± 0.74	4.23 ± 5.43
II	Cataluña	11	73.79 ± 3.72	45.65 ± 1.10	3.83 ± 0.14	5.09 ± 0.76
		20	66.86 ± 2.63	37.55 ± 0.96	3.59 ± 0.07	4.22 ± 0.18
		Mean And DS	70.33 ± 4.90	41.60 ± 5.73	3.71 ± 0.17	4.66 ± 0.62
III	Comunidad Foral Navarra	4	82.52 ± 0.85	78.54 ± 0.80	4.94 ± 0.05	5.71 ± 1.07
		12	65.73 ± 1.60	40.02 ± 0.82	4.85 ± 0.09	4.68 ± 1.34
		Mean And DS	74.13 ± 11.87	59.28 ± 27.24	4.90 ± 0.06	5.20 ± 0.73
IV	Galicia	5	67.46 ± 3.05	39.32 ± 1.52	4.06 ± 0.05	5.20 ± 1.02
		7	71.56 ± 4.41	42.91 ± 1.18	4.11 ± 0.02	4.00 ± 0.14
		8	62.20 ± 3.22	41.94 ± 0.93	3.68 ± 0.02	4.46 ± 0.16
		9	73.22 ± 1.54	69.50 ± 0.96	4.06 ± 0.04	5.06 ± 0.82
		10	71.63 ± 9.17	39.88 ± 2.21	3.81 ± 0.14	5.48 ± 0.61
		16	48.20 ± 0.99	26.65 ± 0.72	1.64 ± 0.04	2.54 ± 1.28
		17	48.50 ± 4.51	29.00 ± 1.93	1.94 ± 0.31	3.30 ± 1.19
		Mean And DS	63.25 ± 2.00	41.31 ± 2.28	3.33 ± 0.19	4.29 ± 0.37
	GLOBAL VALUES		65.74 ± 8.89	42.72 ± 13.19	3.86 ± 0.81	4.36 ± 0.68

**Table 2 microorganisms-11-01429-t002:** Antibacterial activity of EEP using disc diffusion method.

Strain	*L. grayii*	*L. innocua*		*L. monocytogenes*							
Diameter Inhibition	L931	L30	L910	L10	L51	L52	L74	L75	L934	L935	L4032
maximum (mm)	28	25	27	38	26	28	26	28	25	24	30
minimum (mm)	14	10	5	10	11	12	12	12	14	11	12

**Table 3 microorganisms-11-01429-t003:** Antibacterial activity measured in MICs (µg/mL) and MBCs (µg/mL) after 48 h of EEP against the strains of *L. grayii*, *L. innocua*, and *L. monocytogenes* in three different pH conditions.

Zone Code		*L. grayii*					*L. innocua*						*L. monocytogenes*											
		L931	L30		L910		L10		L51		L52		L74		L75		L934		L935		L4032			
	pH	MIC ^b,c^	MBC	MIC ^b,c^	MBC	MIC ^b,c^	MBC	MIC ^a,c^	MBC	MIC ^b,c^	MBC	MIC ^b,c^	MBC	MIC ^b,c^	MBC	MIC ^b,c^	MBC	MIC ^b,c^	MBC	MIC ^b,c^	MBC	MIC ^b,c^	MBC	MIC Means
I	7.04	269.53	252.52	197.27	428.24	263.67	413.77	250.00	532.41	189.45	259.69	185.55	384.84	220.70	321.44	197.27	335.76	210.94	308.16	250.00	312.5	234.38	410.88	270.46
	6.01	183.67	225.13	188.48	221.35	175.63	193.92	194.34	421.01	222.66	227.74	195.31	303.31	222.66	295.14	207.03	208.21	161.28	215.57	154.97	209.78	189.98	257.52	209.15
	5.01	119.14	154.30	121.09	127.97	143.03	190.25	130.86	224.25	161.13	211.95	154.30	186.69	160.16	184.70	167.97	168.95	152.57	161.06	115.23	137.35	125.00	147.16	144.53
II	7.04	425.00	468.75	468.75	937.50	468.75	937.50	468.75	781.25	312.50	468.75	468.75	625.00	468.75	625.00	625.00	625.00	390.63	468.75	468.75	625.00	468.75	468.75	476.56
	6.01	321.50	395.31	234.38	312.50	321.50	395.31	234.38	312.50	273.44	332.03	312.50	468.75	390.63	434.38	390.63	412.50	223.88	234.38	238.88	390.63	280.19	234.38	300.25
	5.01	238.88	390.63	156.25	312.50	197.56	234.38	195.31	312.50	234.38	234.38	234.38	312.50	234.38	468.75	234.38	368.75	197.56	234.38	197.56	390.63	234.38	234.38	216.42
III	7.04	156.25	217.19	234.38	312.50	97.66	195.31	234.38	312.50	312.50	312.50	312.50	312.50	195.31	234.38	234.38	312.50	117.19	234.38	156.25	156.25	156.25	234.38	197.27
	6.01	138.88	197.66	156.25	234.38	238.88	275.78	195.31	234.38	156.25	468.75	156.25	390.63	156.25	195.31	156.25	390.63	197.56	198.31	219.34	234.38	138.88	234.38	191.48
	5.01	78.13	117.19	117.19	117.19	58.59	117.19	156.25	156.25	78.13	78.13	78.13	78.13	117.19	117.19	117.19	156.25	58.59	117.19	58.59	78.13	58.59	178.13	97.66
IV	7.04	128.35	245.54	239.96	379.46	122.77	357.14	206.47	234.38	345.98	368.30	167.41	412.95	167.41	245.54	212.05	334.82	122.77	290.18	156.25	223.21	172.99	267.86	183.59
	6.01	156.89	491.07	103.24	167.41	146.38	161.83	103.24	156.25	178.57	222.77	117.19	290.18	161.83	178.57	200.89	223.21	140.15	234.38	145.73	223.21	192.30	233.93	154.32
	5.01	122.77	200.45	106.03	172.99	167.41	167.41	103.24	122.77	128.35	156.03	117.19	200.89	133.93	136.03	111.61	189.73	100.45	183.71	106.03	118.13	119.98	194.87	123.11
total	7.04	274.67	292.09	294.22	241.97	289.02	289.67	297.43	296.28	232.47	266.26	253.23	267.16	253.49	274.89	289.02	296.28	192.34	266.26	276.04	288.89	250.19	274.89	n/a
	6.01	229.78	166.41	165.31	212.09	212.92	212.51	173.29	174.00	190.57	183.02	233.14	182.29	233.35	178.55	212.92	174.00	175.06	183.02	178.57	212.16	229.85	178.55	n/a
	5.01	132.21	134.12	102.96	132.25	156.67	158.13	133.42	133.85	142.65	137.08	141.32	142.90	153.18	112.45	156.67	133.85	123.92	137.08	112.12	139.05	129.47	112.45	n/a

^a^ statistically significant difference between pH 7.04 and pH 6.01, ^b^ statistically significant difference between pH 6.01 and pH 5.01, ^c^ statistically significant difference between pH 7.04 and pH 5.01; n/a. not assigned.

## Data Availability

Data will be available upon reasonable request to the corresponding author.

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
