# Peer review of "Antimicrobial Activity of Spanish Propolis against Listeria monocytogenes and Other Listeria Strains"

_microorganisms, 2023, doi:10.3390/microorganisms11061429_

Round 1

Reviewer 1 Report

Dear Authors,

I suggest you some comments before considering the publication of your MS:

After the first time in which you write the name of a microorganism, you can abbreviate the genus by using the first letter. 

Provide the references to all the paragraphs. 

Use the units in their abbreviate form.

Review the proper italics use. 

Use the ABC system to show statistical differences in the tables. 

Why didn't you remove the ethanol from the extract? Being in your sample, maybe you are having a synergistic effect.

Use appropriately the abbreviations.

Can you identify the compounds that have the antimicrobial properties? 

In general, the MS is well-written. Just review the proper use of abbreviations.  in the whole MS.

Reviewer 2 Report

The manuscript describes a topic of great interest to the food industry.

In table 2 explain: How the analysis of the antibacterial activity was carried out. It is not very clear in the table

Were the 31 extracts evaluated?

Was the effectiveness of the extract evaluated by geographic areas?

Are the data extracted from all the 31 extracts evaluated?

There is an inhibition between 5 and 30mm. However, It is important to mention in the text and in the tables, if the differences between these inhibition data can be given by the propolis collection areas.

The author should review the format of table 3.

In the conclusion it would be important to mention which was the propolis that presented the greatest activity against Listeria

Reviewer 3 Report

The manuscript studied different samples of propolis from Spain against Listeria spp. strains. This study contributes in the scientific field, since it is offering a natural alternative to control this important pathogenic microorganism that is commonly found in foods. 

The study was very well designed, performed and presented. Congratulations for the authors.

Table 3 requires adjustments in the format and "ph", can be changed by "pH".

Usually, for the chemical characterization of natural products and bee products, it is interesting to include a HPLC fingerprint in order to better demonstrate the botanical source. So, if the authors possess this information, it could be interesting to include in the manuscript in order to improve its quality. 

Some authors previously published that for natural products be considered "antimicrobial" MIC values could be less than 100 ug/mL and for isolated compounds less than 10 ug/mL. Maybe, could be interesting the authors discuss something about the values found taking it in mind.

Besides this, I consider the information obtained valuable for future developments and uses in the Listeria control. 

Reviewer 4 Report

Manuscript n. 2398241

Article

Antimicrobial activity of Spanish propolis against Listeria

monocytogenes and other Listeria strains

The study is interesting because it seeks environmentally friendly solutions to control contamination in food that is very dangerous to human health. However, I note a major problem related to the characteristic flavour of propolis and the fact that this flavour would, in my opinion, alter the flavour of the food. This aspect does not seem to me to have been well discussed.

Another aspect overlooked in my opinion is that if propolis is obtained in an area rich in plants that have known antibacterial properties (e.g. thyme), the effect could be 'plant-dependent'. Can this aspect be discussed further?

Other points of attention are formal:

Check address and affiliation

Check English

Check Acknowledgements

 Moderate editing of English language
